# LEARNING PREDICTIVE COMMUNICATION BY IMAGINATION IN NETWORKED SYSTEM CONTROL

## ABSTRACT

Dealing with multi-agent control in networked systems is one of the biggest challenges in Reinforcement Learning (RL) and limited success has been presented compared to recent deep reinforcement learning in single-agent domain. However, obstacles remain in addressing the delayed global information where each agent learns a decentralized control policy based on local observations and messages from connected neighbors. This paper first considers delayed global information sharing by combining the delayed global information and latent imagination of farsighted states in differentiable communication. Our model allows an agent to imagine its future states and communicate that with its neighbors. The predictive message sent to the connected neighbors reduces the delay in global information. On the tasks of networked multi-agent traffic control, experimental results show that our model helps stabilize the training of each local agent and outperforms existing algorithms for networked system control.

## 1 INTRODUCTION

Networked system control (NSC) is extensively studied and widely applied, including connected vehicle control (Jin & Orosz, 2014), traffic signal control (Chu et al., 2020b), distributed sensing (Xu et al., 2016), networked storage operation (Qin et al., 2015) etc. In NSC, agents are connected via a communication network for a cooperative control objective. For example, in an adaptive traffic signal control system, each traffic light performs decentralized control based on its local observations and messages from connected neighbors. Although deep reinforcement learning has been successfully applied to some complex problems, such as Go (Silver et al., 2016), and Starcraft II (Vinyals et al., 2019), it is still not scalable in many real-world networked control problems. Multi-agent reinforcement learning (MARL) addresses the issue of scalability by performing decentralized control. Recent decentralized MARL performs decentralized control based on the assumptions of global observations and local or global rewards (Zhang et al., 2018; 2019a; Qu et al., 2019; 2020b;a), which are reasonable in multi-agent gaming but not suitable in NSC. A practical solution is to allow each agent to perform decentralized control based on its local observations and messages from the connected neighbors. Various communication-based methods are proposed to stabilize training and improve observability, and communication is studied to enable agents to behave as a group, rather than a collection of individuals (Sukhbaatar & Fergus, 2016; Chu et al., 2020a).

Despite recent advances in neural communication (Sukhbaatar & Fergus, 2016; Foerster et al., 2016; Chu et al., 2020a), delayed global information sharing remains an open problem that widely exists in many NSC applications. Communication protocol not only reflects the situation at hand but also guides the policy optimization. Recent deep neural models (Sukhbaatar & Fergus, 2016; Foerster et al., 2016; Hoshen, 2017) implement differentiable communication based on available connections. However, in NSC, such as traffic signal control, each agent only connects to its neighbors, leading to a delay in receiving messages from the distant agents in the system, and the non-stationarity mainly comes from these partial observation (Chu et al., 2020a). Communication with delayed global information limits the learnability of RL because RL agents can only use the delayed information and not leverage potential future information. Moreover, it is not efficient in situations where an environment is sensitive when the behaviours of agents change. It is therefore of great practical relevance to develop algorithms which can learn beyond the communication with the delayed information sharing.

In this paper we introduce ImagComm that learns communication by imagination for multi-agent reinforcement learning in NSC. We leverage the model of the agent's world to provide an estimate of farsighted information in latent space for communication. At each time step, the agent is allowed to imagine its future states in an abstract space and convey this information to its neighbors. Therefore unlike previous works, our communication protocol conveys not only the current sharing information but also the *imagined* sharing information. It is applicable whenever communication changes frequently, e.g. at every time step agents may receive new communication information.

We summarize our main contributions as follows: (1) We first introduce the imagination module that can be used to learn latent dynamics for communication in networked multi-agent systems control. (2) We predict the future state of each local agent and allow each agent to convey the latent state to neighbors as messages, which reduce the delay of global information. (3) We demonstrate that leveraging the predictive communication by imagination in latent space succeeds in networked system control. We explore this model on a range of NSC tasks. Our results demonstrate that our method consistently outperform baselines on these tasks.

## 2 RELATED WORK

Networked system control (NSC) considers the problem where agents are connected via a communication network for a cooperative control objective, such as autonomous vehicle control (Jin & Orosz, 2014), adaptive traffic signal control (Chu et al., 2020b), and distributed sensing (Xu et al., 2016), etc. Recently reinforcement learning has become popular for NSC through decentralized control and communications by networked agents.

Communication is an important part for multi-agent RL to compensate for the information loss in partial observations. Heuristic communication allows the agents to share some certain forms of information, such as policy fingerprints from other agents (Foerster et al., 2017) and averaged neighbor's policies (Yang et al., 2018). Recently end-to-end differentiable communications have become popular (Foerster et al., 2016; Sukhbaatar & Fergus, 2016; Chu et al., 2020a) since the communication channel is learned to optimize the performance. Attention-based communication (Hoshen, 2017; Das et al., 2019; Singh et al., 2019) selectively send messages to the agents chosen, however, these are not suitable for NSC since the communication is allowed only between connected neighbors. Our method adopts differentiable communication with end-to-end training. Compared to existing works, we introduce a new predictive communication module through learning latent dynamics.

Learning latent dynamics has been studied to solve single agent tasks, such as E2C (Watter et al., 2015), RCE (Banijamali et al., 2018), PlaNet (Hafner et al., 2019), SOLAR (Zhang et al., 2019b) and so on. Lee et al. (2019) and Gregor et al. (2019) learn belief representations to accelerate model-free agents. World Models (Ha & Schmidhuber, 2018) learn latent dynamics in a two-stage process to evolve linear controllers in imagination. I2A (Racanière et al., 2017) hands imagined trajectories to a model-free policy based on a rollout encoder. In contrast to these works, our work considers multi-agent tasks and learns predictive communication by imagination in latent space.

## 3 PRELIMINARIES

In networked system control problem, we work with a networked system, which is described by a graph $G(\mathcal{V}, \mathcal{E})$, where $i \in \mathcal{V}$ denotes the $i$th agent and $ij \in \mathcal{E}$ denotes the communication link between agents $i$ and $j$. The corresponding networked (cooperative) multi-agent MDP is defined by a tuple $(G, \{\mathcal{S}_i, \mathcal{A}_i\}_{i \in \mathcal{V}}, \{\mathcal{M}_{ij}\}_{ij \in \mathcal{E}}, p, \{r_i\}_{i \in \mathcal{V}})$. $\mathcal{S}_i$ and $\mathcal{A}_i$ are the local state space and action space of agent $i$. Let $\mathcal{S} := \cup_{i \in \mathcal{V}} \mathcal{S}_i$ and $\mathcal{A} := \cup_{i \in \mathcal{V}} \mathcal{A}_i$, the MDP transitions follow a stationary probability distribution $p : \mathcal{S} \times \mathcal{A} \times \mathcal{S} \to [0, 1]$. The global reward is denoted by $r : \mathcal{S} \times \mathcal{A} \to \mathbb{R}$ and defined as $r = \frac{1}{|\mathcal{V}|} \sum_{i \in \mathcal{V}} r_i$ indicating that all local rewards are shared globally. The communication is limited to neighborhoods. $\mathcal{M}$ denotes the message space for the communication model. That is each agent $i$ observes $\tilde{s}_{i,t} := s_{i,t} \cup m_{\mathcal{N}_i i, t}$, where $s_{i,t} \in \mathcal{S}_i$ denotes local state space of agent $i$ and $m_{\mathcal{N}_i i, t} := \{m_{ji,t}\}_{j \in \mathcal{N}_i}$ and $\mathcal{N}_i := \{j \in \mathcal{V} | ji \in \mathcal{E}\}$. Message $m_{ji,t} \in \mathcal{M}_{ji}$ denotes all the available information at an agent's neighbor. In NSC, the system is decentralized and the communication is limited to neighborhoods. Each agent $i$ follows a decentralized policy $\pi_i : \tilde{\mathcal{S}}_i \times \mathcal{A}_i \to [0, 1]$

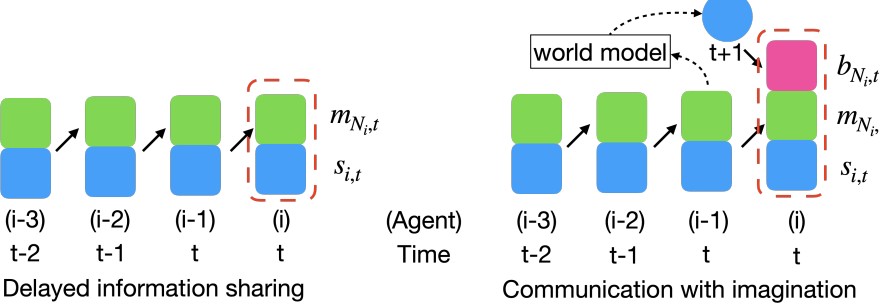

Figure 1: Flow diagram of predictive communication. **Left:** An illustration of a networked system example is an adaptive cruise control (ACC) system with vehicle-to-vehicle (V2V) communication. For the information sharing, agent $i$ knows information at time $t$ including $s_{i,t} \cup s_{i-1,t} \cup s_{i-2,t-1} \cup s_{i-3,t-2}$, which are the delayed global observations. For the communication, based on world models $b_{i,t}$ is learned from its neighbor's future states, which compensate for delayed information. **Right:** policy representation that includes messages of neighbors generated by the imagined module.

to choose its own action $a_{i,t} \sim \pi_i(\cdot|\tilde{s}_{i,t})$ at time $t$. The objective is to maximize $\mathbb{E}_\pi[R_0]$, where $R_t = \sum_{l=0}^\infty \gamma^l r^{t+l}$ and $\gamma$ is a discount factor.

## 4 METHODOLOGY

Our goal is to learn predictive communication on a particular observation or environment state. We start by introducing the networked MDP with neighborhood communications and delayed information issue in communication. Then, we describe ImagComm that utilizes predictive communication, which we learn the agent's world model to provide an additional context for communication.

### 4.1 DELAYED COMMUNICATION IN NETWORKED SYSTEM CONTROL

Following the setting of NSC in (Chu et al., 2020a), we assume that all messages sent from agent $i$ are identical and we denote $m_{ij} = m_i, \forall j \in \mathcal{N}_i$. The message explicitly includes state $s$ and policy $\pi$ and agent belief $h$, i.e., $m_{i,t} = s_{i,t} \cup \pi_{i,t-1} \cup h_{i,t-1}$ in communication. Note that $\pi_{i,t-1}$ is the probability distribution over discrete actions. Thus for each agent in NSC, $\tilde{s}_{i,t} := s_{\mathcal{V}_{i,t}} \cup \pi_{\mathcal{N}_i,t-1} \cup h_{\mathcal{N}_i,t-1}$.

Note the communication phase is prior-decision, so only $h_{i,t-1}$ and $\pi_{i,t-1}$ are available. This protocol can be easily extended for multi-pass communication. We assume that any information that agent $j$ knows at time $t$ can be included in $m_{ji,t}$ and $m_{ji,t} = s_{j,t} \cup \{m_{kj,t-1}\}_{k \in \mathcal{N}_j}$. Then $\tilde{s}_{i,t} := s_{i,t} \cup \{s_{j,t+1-d_{ij}}\}_{j \in \mathcal{V}/\{i\}}$, which includes the delayed global observations. $d_{ij}$ indicates the distance between $i$ and $j$, i.e. the hops between two agents on the graph of the networked system. We illustrate the delayed information in Figure 1. A more rigorous analysis of this conclusion can be found in the Appendix A.

### 4.2 PREDICTIVE COMMUNICATION

To reduce the delay of global information, we consider a forward model for predicting future states of each agent $j$, then $s_{j,t+1}$ can be encoded as a message for communication, and agent $i$ can benefit from this information. Let $\hat{s}_{i,t}$ be the abstract state of $i$th agent, $W_i \in \mathcal{W}_i$ be a world model of the transition dynamics from $\hat{s}_{i,t}$ to the abstract state $\hat{s}_{i,t+1}$, and let $b_{i,t} := \cup_{\tau=1}^k \hat{s}_{i,t+\tau}$ denote the predictive message. We aim to build a policy based on delayed global observations and predictive messages. The value of policy $\pi_i$ can be defined as $V_i^{\pi,W}(s)$ based on the model $W_i$:

$$V_i^{\pi,W}(s, a_{\mathcal{N}_i}) = \mathbb{E}_{a_{i,t} \sim \pi_i(\cdot|\tilde{s}_{i,t}, b_{i,t})}\big[R_{i,t}^\pi \mid \tilde{s}_t = s, a_{\mathcal{N}_i,t} = a_{\mathcal{N}_i}\big]. \tag{1}$$

Learning based on (1) has the benefit of reduced delay in global information compared to that without $b_{i,t}$; this is formally presented in Proposition 1. Proofs are provided in Appendix A.

**Proposition 1.** *ImagComm can reduce the delay of global information by incorporating a predictive model in the communication protocol.*

We are now interested in constructing an abstract model $\widehat{W}_i(\cdot; \varphi)$ to approximate $W_i$, which operates on an abstract state. Let $\hat{s}_{i,t+1}$ be the new abstract state sampled by $\hat{s}_{i,t+1} \sim \widehat{W}_i(\hat{s}_{i,t})$. We want to minimize $\|\hat{s}_{i,t+1} - g_i(s_{i,t+1}; \psi)\|$, where $g_i(\cdot; \psi)$ is an embedding of raw states. Let $V_i^{\pi, \widehat{W}}$ be the value function of the policy on the estimated model $\widehat{W}_i$. Towards optimizing $V_i^{\pi, W^\star}(s, a_{\mathcal{N}_i})$, we build a lower bound as follows and maximize it iteratively:

$$V_i^{\pi, W^\star}(s, a_{\mathcal{N}_i}) \geq V_i^{\pi, \widehat{W}}(s, a_{\mathcal{N}_i}) - D(\widehat{W}, \pi), \tag{2}$$

where $D(\widehat{W}, \pi) \in \mathbb{R}$ bounds the discrepancy between $V_i^{\pi, W^\star}$ and $V_i^{\pi, \widehat{W}}$. In practice, $D(\widehat{W}_i, \pi_i)$ is defined as

$$D_{\pi_i^{\mathrm{ref}}}(\widehat{W}_i, \pi_i) = \alpha \cdot \mathbb{E}_{s_0, \ldots, s_t, \sim \pi_i^{\mathrm{ref}}}[\|\widehat{W}_i(\hat{s}_{i,t}) - g_i(s_{i,t+1})\|], \tag{3}$$

where $\alpha$ is a hyperparameter, $\pi_i^{\mathrm{ref}}$ is the policy used for sampling. For each agent, we solve the following problem:

$$\pi^{k+1}, W^{k+1} = \operatorname*{argmax}_{\pi \in \Pi, W \in \mathcal{W}} \quad V_i^{\pi, W} - D_{\pi_i^k, \delta}(W, \pi). \tag{4}$$

With the predictive imagination module, each agent utilizes the estimate of predictive state information to learn its belief and optimize the control performance of all other agents. Follow the analysis in (Luo et al., 2018), we can show that ImagComm can lead to monotonic improvement in policy iteration. Proofs are defered to Appendix A.

**Proposition 2.** *Suppose that $W_i^* \in \mathcal{W}_i$ is the optimal model and the optimization problem in equation (4) is solvable at each iteration. Solving (4) produces a sequence of policies $\pi_i^0, \ldots, \pi_i^T$ with monotonically increasing values:* $V_i^{\pi^0, W^*} \leq V_i^{\pi^1, W^*} \leq \cdots \leq V_i^{\pi^T, W^*}$.

A conclusion following directly from Proposition 2 is that solving (4) will converge to a local maximum. ImagComm considers build a world model and predict the farsighted state by a imagination module to eliminate the delay in global information and henceforth reduce the negative influence of the partial observability. Because the future information after time $t$ compensate for some of the delayed information at time $t$. Next we will present the differentiable neural communication with imagination.

### 4.3 DIFFERENTIABLE NEURAL COMMUNICATION

In our approach, an agent performs the following operations throughout the agent's life time: learning the latent dynamics model from the dataset of past experience to predict future states of itself, encoding the imagined features into message and learning differentiable communication models together with the predictive information. Specifically, the predictive model lets us predict the states ahead in the latent space without having to observe.

Different with previous works that use $h_{i,t} = g_{\mathcal{V}_i}(h_{i,t-1}, s_{i,t}, m_{\mathcal{N}_i,t})$, we propose to learn communication with imagination (ImagComm), as shown in Figure 1, to add the *imagined* delayed information to communication:

$$h_{i,t} = g_{\mathcal{V}_i}(h_{i,t-1}, s_{i,t}, m_{\mathcal{N}_i,t}, b_{\mathcal{N}_i,t}) \tag{5}$$

where $b_{\mathcal{N}_i,t}$ indicates the module of the predictive message, $m_{\mathcal{N}_i,t}$ represents the standard communication module, which is the same as previous communication work. $g_{\mathcal{V}_i}$ is a differentiable function to extract informaton for the agent's beliefs. For example, $g_{\mathcal{V}_i}$ can be LSTM (Hochreiter & Schmidhuber, 1997).

Compared to (Chu et al., 2020a), ImagComm uses imagination module to provide an augmented observation of agents and reduces delays in global information. Compared to model-based approaches (Luo et al., 2018; Janner et al., 2019), the differences are two-fold: i) instead of learning a model for a single-agent MDP, each agent learns a decentralized predictive model locally; ii) instead of aiming to improve the sampling efficiency, we aim to augment the message for communication, thus reducing the delay in global information from each agent's view; this can be viewed as a combination of model-based and model-free aspects.

### 4.4 PRACTICAL IMPLEMENTATIONS

Simply we assume each agent is based on A2C (Advantage Actor-Critic) models. Let $\{\pi_{\theta_i}\}_{i\in\mathcal{V}}$ and $\{V_{\omega_i}\}_{i\in\mathcal{V}}$ be the decentralized actor-critics, and $\{(s_{i,\tau}, m_{\mathcal{N}_i i,\tau}, a_{i,\tau}, r_{i,\tau}, b_{\mathcal{N}_i i,\tau})\}_{i\in\mathcal{V},\tau\in\mathcal{B}}$ be the on-policy minibatch from networked MDPs under stationary policies $\{\pi_{\theta_i}\}_{i\in\mathcal{V}}$. For each agent with the belief $h_{i,t}$, the actor and critic become $\pi_{\theta_i}(h_i)$ and $V_{\omega_i}(h_i, a_{\mathcal{N}_i})$ for fitting the optimal policy $\pi_i^*$ and value function $V^{\pi_i}$. ImagComm has three components based on (5). $m_{\mathcal{N}_i,t}$ represents the message that will be passed to $h_{i,t}$ with $m_{\mathcal{N}_i,t} = s_{\mathcal{N}_i,t} \cup \pi_{\mathcal{N}_i,t-1} \cup h_{\mathcal{N}_i,t-1}$. For the predictive message $b_{i,t}$, $\hat{s}_{i,t} = f_i(s_{\mathcal{V}_i,t}, h_{i,t-1}; \phi_i)$, and $\hat{s}_{i,t+1} = \widehat{W}_i(\hat{s}_{i,t})$. Let $\Phi_i = \{\phi_i, \psi_i, \varphi_i\}$ denote the parameters for abstract state embedding, raw state embedding, and the imagination module for agent $i$. Then each actor and critic are updated by losses

$$\mathcal{L}(\theta_i) = \frac{1}{|\mathcal{B}|}\sum_{\tau\in\mathcal{B}}\left(-\log\pi_{\theta_i}(a_{i,\tau}\mid h_{i,\tau})\right)A_{i,\tau}^{\pi} + \beta H[\pi_{\theta_i}(a_i\mid h_{i,\tau})], \tag{6a}$$

$$\mathcal{L}(\omega_i) = \frac{1}{|\mathcal{B}|}\sum_{\tau\in\mathcal{B}}\left(R_{i,\tau}^{\pi} - V_{\omega_i}(h_{i,\tau}, a_{\mathcal{N}_i,\tau})\right)^2, \tag{6b}$$

$$\mathcal{L}(\Phi_i) = \frac{\alpha}{|\mathcal{B}|}\sum_{\tau\in\mathcal{B}}\left[\|\widehat{W}_i(f_i(s_{\mathcal{V}_i,\tau}, h_{i,\tau-1})) - g_i(s_{i,\tau+1})\|\right], \tag{6c}$$

where $\hat{R}_{i,\tau}^{\pi} = \sum_{\tau'=\tau}^{\tau_B-1}\gamma^{\tau'-\tau}r_{i,\tau'} + \gamma^{\tau_B-\tau}v_{i,\tau_B}$ is the target action-value, $v_{i,\tau} = V_{\omega_i-}(\tilde{s}_{i,\tau}, a_{\mathcal{N}_i,\tau})$ is the state-value as baseline, $A_{i,\tau}^{\pi} = R_{i,\tau}^{\pi} - v_{i,\tau}$ is the advantage function as critic, and $\beta$ is the hyperparameter of the entropy loss. In implementation of Eq. (6c), we adopt the root mean square loss (RMSE). The overall algorithm is provided in Algorithm 1.

---

**Algorithm 1** Imagination-based policy optimization

---

1: Initialize policy $\{\pi_i^0\}_{i\in\mathcal{V}}$, predictive model $\{\widehat{W}_i^0\}_{i\in\mathcal{V}}$, empty mini-batch $\mathcal{D}_{\mathcal{B}}$.
2: **for** k = 0, 1,..., T **do**
3:     Collect a batch of data with $\{\pi_i^k\}_{i\in\mathcal{V}}$ in real environment:
        $\mathcal{D}_{\mathcal{B}} = \{(s_{i,\tau}, m_{\mathcal{N}_i i,\tau}, a_{i,\tau}, r_{i,\tau}, b_{\mathcal{N}_i i,\tau})\}_{i\in\mathcal{V},\tau\in\mathcal{B}}$ .
4:     Update policy $\{\pi_i^{k+1}\}_{i\in\mathcal{V}}$ under imagination by solving (6a).
5:     Update critic $\{V_i^{k+1}\}_{i\in\mathcal{V}}$ under imagination by solving (6b).
6:     Update $f_i^{k+1}, g_i^{k+1}$ and $\widehat{W}_i^{k+1}$ on dataset $\mathcal{D}_{\mathcal{B}}$ by solving (6c).
7: **end for**

---

## 5 NUMERICAL EXPERIMENTS

We evaluate ImagComm on several challenging environments in networked system control and compare it to current state-of-the-art algorithms for communication.

### 5.1 ENVIRONMENTS

We use four existing simulation environments: ATSC Grid, ATSC Monaco, Cooperative Adaptive Cruise Control (CACC) Catch-up and CACC Slow-down (Chu et al., 2020b). The ATSC environments are developed based on SUMO (Krajzewicz et al., 2012). In **ATSC**, ATSC Grid simulates a $5 \times 5$ synthetic traffic grid , as shown in Figure 2a. ATSC Monaco simulates a real-world 28-intersection traffic network from Monaco city, as shown in Figure 2b. In homogeneous scenario, i.e. ATSC Grid, all agents have the same action space consisting of five pre-defined signal phases. While in heterogeneous scenario, i.e. ATSC Monaco, agents have a variety of action spaces. For both scenarios, The objective of ATSC is to adaptively control traffic lights at the intersections to minimize traffic congestion based on real-time road-traffic measurements. Local state is defined as $s_{t,i} = \{\text{wait}_t[l], \text{wave}_t[l]\}_{ji\in\mathcal{E},l\in L_{ji}}$, where $l$ is each incoming lane of intersection $i$. Wait[·] measures the cumulative delay of the first vehicle and wave[·] measures the total number of approaching vehicles along each incoming lane within 50m to the intersection. Rewards for each agent are defined as $r_{i,t} = -\sum_{ji\in\mathcal{E},l\in L_{ji}}\left(\text{queue}_{t+\Delta t}[l]\right)$, where queue[·] denotes the number of queuing vehicles on an approaching lane, which is measured by induction-loop detectors (ILD). In

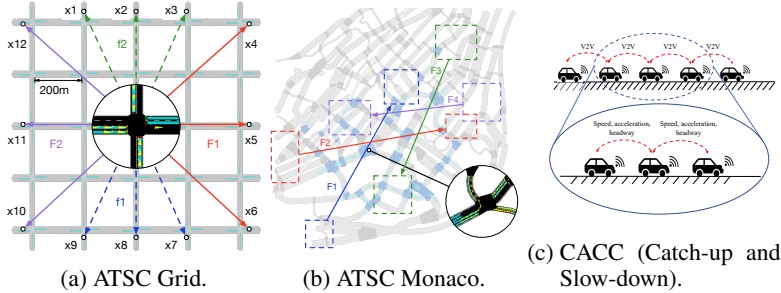

(a) ATSC Grid.  (b) ATSC Monaco.  (c) CACC (Catch-up and Slow-down).

Figure 2: Environments on adaptive traffic signal control (ATSC) and cooperative adaptive cruise control (CACC) systems.

**CACC**, CACC Catch-up scenario simulates a string of $8$ vehicles for 60s with a $0.1$s control interval, where target speed $v_t^* = 15m/s$ and initial headway $h_{1,0} > h_{i,0}, \forall i \neq 1$. CACC Slow-down also simulates a string of $8$ vehicles for 60s with a $0.1$s control interval, where initial headway $h_{i,0} = h^*$ and target speed $v_t^*$ linearly decreases to 15m/s during the first 30s and then stays at constant. In both scenarios, the objective is to adaptively coordinate a platoon of vehicles to minimize the car-following headway and speed perturbations based on real-time vehicle-to-vehicle communication. Each vehicle observes and shares its headway $h$, velocity $v$, and acceleration $a$ to neighbors within one step. Models are trained to recommend appropriate hyper-parameters $(\alpha^\circ, \beta^\circ)$ for each OVM controller(Bando et al., 1995), selected from four levels $\{(0,0),(0.5,0),(0,0.5),(0.5,0.5)\}$. Rewards are designed as a cost function. Assuming the target headway and velocity profile are $h^* = 20m$ and $v_t^*$, respectively, the cost of each agent is $(h_{i,t} - h^*)^2 + (v_{i,t} - v_t^*)^2 + 0.1u_{i,t}^2$. Whenever a collision happens ($h_{i,t} < 1m$), a large penalty of 1000 is assigned to each agent and the state becomes absorbing. An additional cost $5\left(2h_{st} - h_{i,t}\right)_+^2$ is provided in training for potential collisions.

## 5.2 BASELINES AND SETUP

All the baselines are implemented based on the A2C agent (Mnih et al., 2016) following the methods in Eq. (6a)(6b). The baselines include one non-communicative policy IA2C (Mnih et al., 2016), and three communicative policies NeurComm (Chu et al., 2020a), CommNet (Sukhbaatar & Fergus, 2016), DIAL (Foerster et al., 2016). The details will be provided in Appendix B.

For ImagComm, we found that two-hop information already lead to compelling performance and we use one-step imagination. We use $h_{i,t} = \text{LSTM}(h_{i,t-1}, \text{concat}(\text{relu}(s_{\mathcal{V}_i,t}), \text{relu}(\pi_{\mathcal{N}_i,t-1}), \text{relu}(h_{\mathcal{N}_i,t-1}), \text{relu}(b_{\mathcal{N}_i,t})))$, and $\hat{s}_{i,t+1} = \tanh(\text{concat}(\text{relu}(s_{\mathcal{V}_i,t}), h_{i,t-1}))$. Then $h_{i,t}$ is fed into two fully-connected neural networks producing policy and value separately. Note that for state encoding and $g_i(\cdot)$, we use one fully-connected layer with $\tanh$ activation. For message extracting $g_{\mathcal{V}_i}$, we use a LSTM layer. All layers are set up with 64 hidden units. For the imagination module, we stack two fully-connected layers. Other settings include actor learning rate $5 \times 10^{-4}$, critic learning rate $2.5 \times 10^{-4}$, entropy coefficient $\beta = 0.01$, batch size $|\mathcal{B}| = 120$. Each method is trained over 1M steps under an actor-critic framework. In ATSC, entropy coefficient $\beta = 0.01$, batch size $|\mathcal{B}| = 120$ and in CACC, $\beta = 0.05$, batch size $|\mathcal{B}| = 60$. We use a different random seed to initialize the environment without loss of generality. As different initial seeds may cause fluctuation, we smooth the learning curve using moving averages with a window size of 100 episodes following NeurComm(Chu et al., 2020a).

## 5.3 TRAINING RESULTS

Figure 3 compares the learning curves of different methods on four environments.The results show that ImagComm overall performs better than others for these environments. In both ATSC environments, our model learns quickly with good stability and performs well in the end. In CACC, the standard deviation of episode returns is high due to the large penalty of collisions. In ATSC Grid, CommNet and DIAL gain little performance improvement even after $0.4$M training steps, while ImagComm and NeurComm learn with a faster speed and end with lower deviation. Imag-

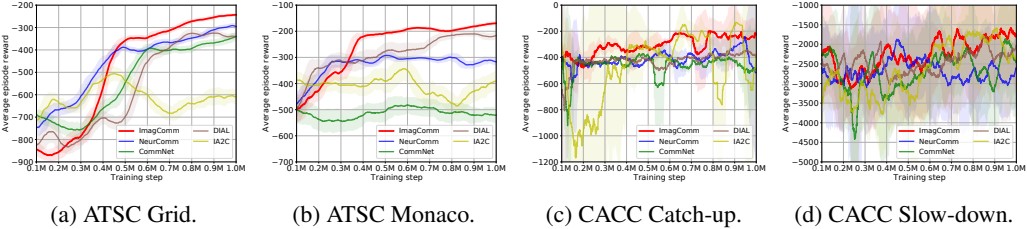

(a) ATSC Grid.  (b) ATSC Monaco.  (c) CACC Catch-up.  (d) CACC Slow-down.

Figure 3: Learning curves for our method and different baselines on four environments

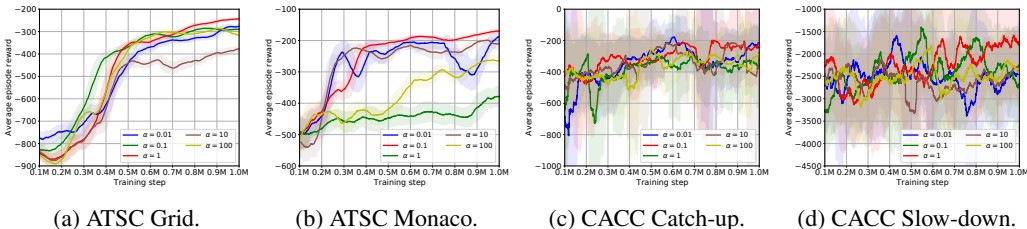

(a) ATSC Grid.  (b) ATSC Monaco.  (c) CACC Catch-up.  (d) CACC Slow-down.

Figure 4: Learning curves for ImagComm with different $\alpha$.

Comm learns slower at the initial state of training owing to the learning of imagination module, but it quickly outperforms NeurComm after 0.5M steps, showing the benefits of incorporating the predictive communication. In ATSC Monaco, DIAL and NeurComm learn fast before 0.3M steps, but NeurComm does not improve much after that, and ImagComm outperforms all the baselines after 0.4M steps. In the CACC environments, ImagComm works better and more stable than others, with a better performance in the end. It shows that adding predictive communication is useful for these tasks. Though IA2C works well sometimes, it is very unstable, which proves the effectiveness of communication. NeurComm and CommNet also work stable in both environments, but suffering from delayed communication in such real-time scenarios, they perform worse than ImagComm.

**Hyper parameter study**  We investigate the impact of coefficient $\alpha$ of $D(\widehat{W}, \pi)$ in our Imag-Comm by comparing the learning curves among $\{0.01, 0.1, 1, 10, 100\}$ on the ATSC and CACC environments in Figure 4. The results show that different $\alpha$ values have different results. In ATSC environments, $\alpha = 1, 0.1$ work best for ATSC Grid and Monaco respectively. In CACC Catch-up scenario, different $\alpha$ values yield similar results. But in CACC Slow-down scenario, $\alpha = 1$ performs best.

## 5.4 EXECUTION PERFORMANCE

Additionally, we investigate the execution results of different methods. We use two more metrics, average queue length and average intersection delay, to get a deeper understanding of controllers' execution impact. Average intersection delay is the mean of waiting time of all vehicles, which is another congestion metric.

Figure 5a and Figure 5b compare average queue length of different trained models in one episode. As shown in the figures, ImagComm achieves the best performance in both scenarios. In ATSC Grid, IA2C and CommNet both fail to reduce traffic pressure and the congestion level remains high in the end. In contrast, ImagComm helps agents communicate with each other effectively, so the queue length increases slower and remains low in the end.

Figure 5c and Figure 5d compare average intersection delay of different trained models in one episode. In Figure 5c, all four communicative policies outperform IA2C dramatically, which proves the effectiveness of communication. However, when it comes to ATSC Monaco, results turn out differently. In Figure 5d, IA2C achieves the lowest average intersection delay while communicative policies perform worse. This can be explained by the emphasis on the queue length as we only consider queue length in reward functions. Intersection delay is not explicitly included in rewards. Thus communicative models may tend to block vehicles in short queues. Although this inclination helps

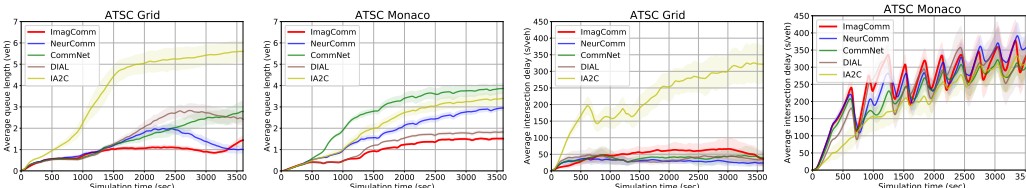

(a) Averaged queue length (b) Averaged queue length (c) Averaged intersection (d) Averaged intersection
in ATSC Grid. in ATSC Monaco. delay in ATSC Grid. delay in ATSC Monaco.

Figure 5: Performance on average queue length (a, b) and average intersection delay (c, d) in the ATSC setting.

Table 1: Average reward comparison over trained models.

| Scenario Name | ImagComm | NeurComm | CommNet | DIAL | IA2C |
|---|---|---|---|---|---|
| ATSC Grid | **-240.91** | -302.68 | -339.16 | -354.52 | -615.85 |
| ATSC Monaco | **-168.18** | -318.90 | -509.26 | -202.91 | -385.71 |
| CACC Catch-up | -196.60 | **-170.27** | -364.16 | -346.85 | -449.27 |
| CACC Slow-down | -1771.99 | -2109.40 | **-1727.24** | -2483.48 | -1732.46 |

Table 2: Performance of MARL controllers in ATSC environments: synthetic traffic grid (top) and Monaco traffic network (bottom). Best values are in bold.

| Temporal Average Metrics | ImagComm | NeurComm | CommNet | DIAL | IA2C |
|---|---|---|---|---|---|
| avg queue length [veh] | **0.88** | 1.17 | 1.47 | 1.64 | 3.84 |
| avg vehicle speed [m/s] | **3.65** | 3.43 | 3.15 | 2.89 | 0.73 |
| trip delay [s] | **194** | 215 | 225 | 262 | 950 |
| avg queue length [veh] | **0.98** | 1.75 | 2.65 | 1.18 | 2.10 |
| avg vehicle speed [m/s] | 0.49 | 0.46 | **0.75** | 0.86 | 1.15 |
| trip delay [s] | 594 | 1445 | 719 | **566** | 479 |

reduce the queue length, average intersection delay becomes longer. This problem can be alleviated through adjusting the reward functions.

We freeze and evaluate our model for another 50 episodes using different seeds and present the average rewards in Table 1. The results show that in ATSC scenarios, ImagComm outperforms other models by a large margin. We further investigate queue length and vehicle speed in ATSC enviroments and present the results in Table 2. In ATSC Grid, ImagComm achieves the lowest queue length and highest average vehicle speed, which means vehicles flow smoothly and ImagComm greatly reduces congestion level. In ATSC Monaco, ImagComm achieves the lowest queue length, but CommNet attains a higher average vehicle speed. This means ImagComm makes vehicles move steadily rather than getting blocked on the road for a long time, but with a sacrifice of moving slowly.

## 6 CONCLUSIONS

In this work, we study the delayed communication problem for decentralized MARL in networked system control. We have introduced an *imagination* module to predict farsighted information for predictive communication. ImagComm combines the delayed global information and predictive state information and performs end-to-end training of the neural communication and imagination module to optimize the control performance in NSC. Extensive empirical studies demonstrate that by leveraging world models for learning the latent delayed information for communication, our method achieves the compelling performance gains in the challenging traffic signal control and adaptive cruise control tasks. We hope that our work will provide inspiration for the research in model-based learning for (networked) multi-agent systems.

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

## A  PROOFS

Chu et al. (2020a) has shown that the communication protocol allows the local agent to utinize the delayed global information. We cite the proof here to stay self-contained.

**Lemma 1.** *(Chu et al., 2020a) By communicating through $h_{i,t} = g_{\mathcal{V}_i}(h_{i,t-1}, s_{i,t}, m_{\mathcal{N}_i,t})$, the delayed global information is utilized to estimate each hidden state, that is*

$$h_{i,t} \supset s_{i,0:t} \cup \left\{ s_{j,0:t+1-d_{ij}}, \pi_{j,0:t-d_{ij}} \right\}_{j \in \mathcal{V} \setminus \{i\}}$$

*where $x \supset y$ if information $y$ is utilized to estimate $x$, and $x_{0:t} := \{x_0, x_1, \ldots, x_t\}$.*

*Proof.* Based on the definition of communication protocol, $m_{i,t} \supset h_{i,t-1}$, and $h_{i,t} \supset h_{i,t-1} \cup s_{\mathcal{V}_{i,t}} \cup \pi_{\mathcal{N}_i,t-1} \cup m_{\mathcal{N}_i,t}$. Hence,

$$
\begin{aligned}
h_{i,t} &\supset s_{i,t} \cup \{s_{j,t}, \pi_{j,t-1}\}_{j \in \mathcal{N}_i} \cup \{h_{j,t-1}\}_{j \in \mathcal{V}_i} \\
&\supset s_{i,t} \cup \{s_{j,t}, \pi_{j,t-1}\}_{j \in \mathcal{N}_i} \cup \left\{ s_{j,t-1} \cup \{s_{k,t-1}, \pi_{k,t-2}\}_{k \in \mathcal{N}_j} \cup \{h_{k,t-2}\}_{k \in \mathcal{V}_j} \right\}_{j \in \mathcal{V}_i} \\
&= s_{i,t-1:t} \cup \{s_{j,t-1:t}, \pi_{j,t-2:t-1}\}_{j \in \mathcal{N}_i} \cup \{s_{j,t-1}, \pi_{j,t-2}\}_{j \in \{\mathcal{V}|d_{ij}=2\}} \cup \{h_{j,t-2}\}_{j \in \{\mathcal{V}|d_{ij}\leq 2\}} \quad (7)\\
&\supset \cdots \\
&\supset s_{i,0:t} \cup \{s_{j,0:t}, \pi_{j,t-2:t-1}\}_{j \in \mathcal{N}_i} \cup \{s_{j,0:t-1}, \pi_{j,0:t-2}\}_{j \in \{V|d_{ij}=2\}} \\
&\quad \cup \ldots \cup \{s_{j,0:t+1-d_{\max}}, \pi_{j,0:t-d_{\max}}\}_{j \in \{\mathcal{V}|d_{ij}=d_{\max}\}},
\end{aligned}
$$

which concludes the proof. $\qquad\square$

### A.1  PROPOSITION 1

*Proof of Proposition 1.* Based on (5), we have that

$$
\begin{aligned}
h_{i,t} &\supset \{m_{j,t}\}_{j \in \mathcal{N}_i} \cup \{b_{j,t}\}_{j \in \mathcal{N}_i} \supset \{h_{j,t-1}\}_{j \in \mathcal{N}_i} \\
&\supset \{m_{j,t-1}\}_{j \in \{\mathcal{V}|d_{ij}=2\}} \cup \{b_{j,t-1}\}_{j \in \{\mathcal{V}|d_{ij}=2\}} \supset \cdots \\
&\supset \{m_{j,t+1-d}\}_{j \in \{\mathcal{V}|d_{ij}=d\}} \cup \{b_{j,t+1-d}\}_{j \in \{\mathcal{V}|d_{ij}=d\}} \supset \cdots
\end{aligned}
\quad (8)
$$

Since $m_{j,t} = s_{j,t} \cup \pi_{j,t-1} \cup h_{j,t-1}$ and $b_{j,t} = \cup_{\tau=1}^k \hat{s}_{j,t+\tau}$ with $k$ the step of forward imagination,

$$h_{i,t} \supset \{s_{j,t+1-d}\}_{j \in \{\mathcal{V}|d_{ij}=d\}} \cup \{\hat{s}_{j,t+k+1-d}\}_{j \in \{\mathcal{V}|d_{ij}=d\}}. \quad (9)$$

Hence, $\hat{s}_{i,t=\tau}$ is included in the observation of agent $j$ at time $\tau + d_{ij} - k - 1$, $s_{i,t=\tau}$, ahead of $s_{i,t=\tau}$ at time $\tau + d_{ij} - 1$ by $k$ steps. $\qquad\square$

### A.2  PROPOSITION 2

*Proof of Proposition 2.* We follow the sketch in (Luo et al., 2018). From (2), we have that $V_i^{\pi_{k+1}, W^*} \geq V_i^{\pi_{k+1}, \widehat{W}} - D_{\pi_k}(\widehat{W}, \pi_{k+1})$. By solving (4), $V_i^{\pi_{k+1}, \widehat{W}} - D_{\pi_k}(\widehat{W}, \pi_{k+1}) \geq V_i^{\pi_k, W^*} - D_{\pi_k}(W^*, \pi_k)$ with $D_{\pi_k}(W^*, \pi_k) = 0$. Thus we have $V_i^{\pi_{k+1}, W^*} \geq V_i^{\pi_k, W^*}$ which completes the proof. $\qquad\square$

## B  EXPERIMENT DETAILS

We describe the baselines used in our experiments as follows:

- NeurComm (Chu et al., 2020a): it is one of the state-of-the-art methods for NSC. It formulates the neighborhood communication under a spatiotemporal MDP and performs independent learning for actors and critics. The messages include the state of current step, policy fingerprints and hidden state of last time step.
- CommNet (Sukhbaatar & Fergus, 2016): it allows agents to communicate through broadcasting a communication vector, which is the average of neighbors' hidden states. The messages include the current state and last step hidden state.

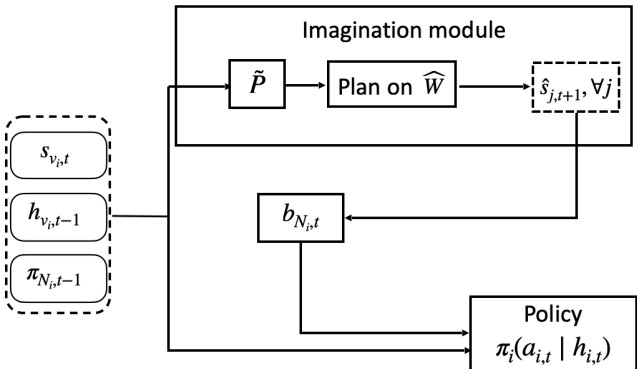

Figure 6: Policy representation that includes messages of neighbors generated by the imagined module.

- DIAL (Foerster et al., 2016): each agent encodes the received messages instead of averaging them, but still sums all encoded inputs. It uses the observations of neighbors as the message.

- IA2C (Mnih et al., 2016): it is an advantage actor-critic method that trains decentralized policies and critics for each agent. Each agent does not communicate with nearby agents. It is implemented similar to MADDPG Lowe et al. (2017) as the critic takes neighboring actions.

- ImagComm: our method adopts an imagination module to predict the delayed message and mitigate nonstationarity coming from partial observations in training. Fig. 6 illustrates the diagram of policy representation.

IA2C is a non-communicative policy while the rest four approaches are communicative policies requiring messages from the neighbors.

In terms of baseline, the communication is based on $h_{i,t} = g_{\mathcal{V}_i}(h_{i,t-1}, s_{i,t}, m_{\mathcal{N}_i,t})$, and the algorithm implementation details are listed below: IA2C: $h_{i,t} = \mathrm{LSTM}(h_{i,t-1}, \mathrm{relu}(s_{i,t}))$. NeurComm: $h_{i,t} = \mathrm{LSTM}(h_{i,t-1}, \mathrm{concat}(\mathrm{relu}(s_{\mathcal{V}_i,t}), \mathrm{relu}(\pi_{\mathcal{N}_i,t-1}), \mathrm{relu}(h_{\mathcal{N}_i,t-1})))$. DIAL: $h_{i,t} = \mathrm{LSTM}(h_{i,t-1}, \mathrm{relu}(s_{\mathcal{V}_t,t}) + \mathrm{relu}(\mathrm{relu}((h_{i,t-1}))) + \mathrm{onehot}(a_{i,t-1})$. CommNet: $h_{i,t} = \mathrm{LSTM}(h_{i,t-1}, \tanh(s_{\mathcal{V}_i,t}) + \mathrm{linear}(\mathrm{mean}(h_{\mathcal{N}_i,t-1})))$. In terms of ImagComm, the communication is based on (5) with $h_{i,t} = \mathrm{LSTM}(h_{i,t-1}, \mathrm{concat}(\mathrm{relu}(s_{\mathcal{V}_i,t}), \mathrm{relu}(\pi_{\mathcal{N}_i,t-1}), \mathrm{relu}(h_{\mathcal{N}_i,t-1}), \mathrm{relu}(b_{\mathcal{N}_i,t})))$. For the imagination module, $\hat{s}_{i,t+1} = \tanh(\mathrm{concat}(\mathrm{relu}(s_{\mathcal{V}_i,t}), h_{i,t-1}))$.

## C  ADDITIONAL RESULTS

Figure 7 shows the convergence of the learning curve for the imagination module. Different from the other three tasks, the ATSC Monaco is tasked with heterogeneous agents in which each agent's state has different dimensions. Thus in Figure 7b we plot the loss that is the summation of the square error of $N$ agents without average normalization by state dimensions. In Figure 7a, 7c and 7d, the y-axis is averaged RMSE loss for $N$ agents.

Figure 8 compares the performance of ImagComm and a modified version of NeurComm-2Hop on CACC task. ImagComm performs one-step prediction to implicitly include the information of two-hop neighbors. NeurComm-2Hop directly uses two-hop information. It is interesting to see that ImagComm outperforms NeurComm-2hop, which proves the effectiveness of the imagination module. Due to the one-step imagination before communication, the receiver obtains the information of the neighbors that are two-hop away and the next-step information of neighbors one-hop away.

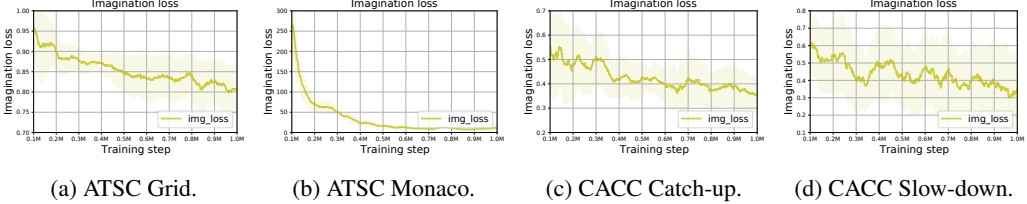

(a) ATSC Grid.  (b) ATSC Monaco.  (c) CACC Catch-up.  (d) CACC Slow-down.

Figure 7: Training curve of the imagination module on ATSC (a, b) and CACC (c, d) setting.

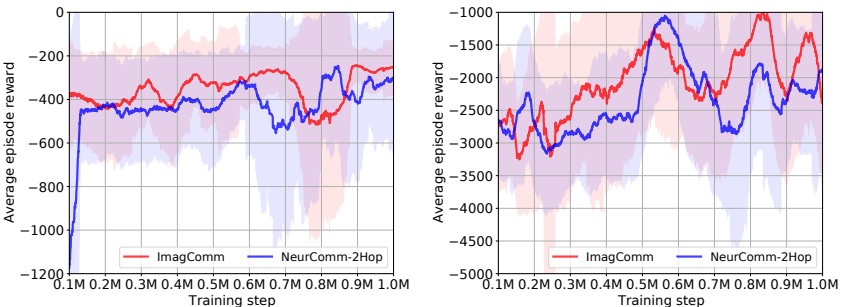

(a) Averaged Reward of CACC Catch-up.  (b) Averaged Reward of CACC Slow-down.

Figure 8: Comparisons between ImagComm and NeurComm-2Hop. ImagComm performs one-step prediction to implicitly include the information of two-hop neighbors. NeurComm-2Hop directly uses two-hop information.

