# OpenReview forum: "Learning Predictive Communication by Imagination in Networked System Control"
_ICLR.cc/2021/Conference — Reject_

### Official Review · AnonReviewer2 · 2020-10-21
**Reviews**

**Rating:** 4
**Confidence:** 2

**Review:**

To reduce the delay of global information, the authors propose an imagination module to predict future information for communication. The agent predicts its future state and shares that with its neighbors. The information delay is an important problem in real-world applications. The proposed ImagComm outperforms other baselines on a range of NSC tasks.

However, I have some concerns:

What are the difference between predicting the future information at the sender end and the receiver end? Since the sender makes a prediction based on its history information, and the history information has been communicated to the receiver, the receiver could make the same prediction using its collected information, which could reduce the communication cost.

$f_i$, $g_i$, and $W_i$ are trained by Eq 6c. The Eq 6c would make the outputs of $W_i$ and $g_i$ close to each other. Would it cause mode collapse, which means that the outputs of $W_i$ and $g_i$ become a constant representation?

In experiments, the ablation study that removes the optimization objective Eq 6c in Algorithm 1 could help to verify that the performance gain is caused by the predictive information rather than the additional information.

---

> ### Author Response · Authors · 2020-11-19
> **Thanks for your constructive comments. We provide a point-wise response below.**
>
> Thanks for your constructive comments. We provide a point-wise response below.
>
> Q: the difference between predicting the future information at the sender end and the receiver end.\
> A: we appreciate your carefulness in noticing this detail. The receiver does not know the states of the sender's neighbor, which were used in prediction of future information. Besides, compared to predicting the future information of $k$ neighbors at each receiver side, the model complexity is reduced by predicting own future states at the sender side.
>
> Q: $f_i$, $G_i$,  and $W_i$ in Eq 6c. \
> A: Firstly, evidenced by the improved performance by including the predictive communication, W_i and g_i do not collapse. Secondly, we have provided  a comparison between ImagComm and a modified NeurComm that uses two-hop information. Please see the updated Fig. 8 in the paper. It is interesting to see that our method outperforms NeurComm with two-hop information, which proves the effectiveness of the imagination module. Due to the one-step imagination before communication, the receiver obtains the information of the neighbors that are two-hop away and the next-step/future information of neighbors that are one-hop away.
>
> Q: In experiments, the ablation study that removes the optimization objective Eq 6c in Algorithm 1 could help to verify that the performance gain is caused by the predictive information rather than the additional information.\
> A: Thanks for the question. If Eq.6c is removed from the optimization,  our method degenerates to NeurComm which is included into comparisons in this paper. Comparing to NeurComm,  ImagComm uses imagination module to provide an augmented observation of agents and outperforms it by a large margin..

---

### Official Review · AnonReviewer1 · 2020-10-26
**Review 1**

**Rating:** 4
**Confidence:** 4

**Review:**

The paper proposes to communicate predicted local states between neighboring agents to address the problem of delayed information in networked multi-agent reinforcement learning. To enable agents to predict future states, a world model is learned at each agent. It is empirically demonstrated that the proposed method has good performance in traffic signal control and cooperative adaptive cruise control.
+ The idea of communicating predicted local states is interesting and could mitigate the problem of delayed information.
+ The proposed method is empirically verified in two scenarios and outperforms other communication methods in MARL.
+ The tuning of hyperparameter (alpha) is also verified by experiments.

Concerns:
- The main concern about this paper is lack of novelty. It seems just applying the world model for single agent directly to MARL without any adaptation. Equations 2, 3, 4 are taken directly from (Luo at el, 2018) without any consideration of the characteristics of MARL. For example, why do these still hold in multi-agent case? Without rigorous mathematical analysis, it is hard to say it is valid. The proofs given in Appendix are not informative at all.
- As in MARL, the state transition depends on the joint action of agents, why does equation 3 hold in practice?  It is just not clear. Any assumption is made for that? I am wondering how well the world model is learned in the experiments.
- In the experiments, only one-step state prediction is used. It is claimed that two-hop information already leads to good performance. However, it is not supported by experiments. What if each agent directly uses two-hop information without prediction (it is easy to implement in simulation). This can serve as a baseline so as to illustrate how the learned model affects the performance empirically.
-  The communication baselines, i.e., CommNet and DIAL, are pretty old. Why not use more recent methods such as ATOC, IC3Net, TarMAC, which perform much better than CommNet and DIAL.

---
**After rebuttal**

The authors' responses do not address my major concerns (the first two). I do not think the responses directly answer my questions.
So, I keep my score unchanged.

---

> ### Author Response · Authors · 2020-11-19
> **Thanks for your detailed comments. Below we provide a point-wise response to your concerns.**
>
> Thanks for your detailed comments. Below we provide a point-wise response to your concerns. We have added the results of world model's convergence and agents using two-hop information in Fig. 7 and Fig. 8.\
> Q: Consideration of the characteristics of MARL. For example, why do these still hold in multi-agent case? Without rigorous mathematical analysis, it is hard to say it is valid. The proofs given in Appendix are not informative at all.\
> A: The value function has been adaptively defined to be consistent with network system control, in which the value of an agent’s policy $\pi$ is related to actions of its neighbors $a_{\mathcal{N}_i}$.
>
> Q: about equation 3 and how the world model is learned in the experiments.\
> A: Eq.(3) is a definition of the quality of the imagination module. Different forms can be considered, but the root mean square error works well for the networked system control problem.  $\hat{s}_{i,t}$ contains the information of neighboring agents defined in line 7 in Sec 4.4.  We provide the convergence of the world model in Fig. 7 in appendix. The world model converges well.
>
> Q: comparisons to agents that directly use two-hop information without prediction.\
> A: Please see the updated Fig. 8 in the paper. we have provided a comparison between ImagComm and a modified NeurComm that uses two-hop information. It is interesting to see that our method outperforms NeurComm with two-hop information, which proves the effectiveness of the imagination module. Due to the one-step imagination before communication, the receiver obtains the information of the neighbors that are two-hop away and the next-step information of neighbors one-hop away.
>
> Q: The communication baselines, i.e., CommNet and DIAL, are pretty old. Why not use more recent methods such as ATOC, IC3Net, TarMAC, which perform much better than CommNet and DIAL.\
> A: ATOC, TarMac and IC3Net are not suitable for networked system control because they learn who to communicate,  while in NSC, agents learn what to communicate based on connections with neighbors on a given network.

---

### Official Review · AnonReviewer3 · 2020-10-27
**Interesting idea but further experimentation is needed to understand the impact of the proposed approach**

**Rating:** 5
**Confidence:** 4

**Review:**

##########################################################################

Summary:

The paper provides an interesting way to add structure to MARL problems that have delay in the communication of state information. By explicitly building a predictive module for the future latent state of the agent and including that predicted state in the passed messages, it is possible that the agent will appropriately pass information that removes the effect of the delay in message passing across the network. They then apply this model to some interesting traffic light and cooperative vehicle control tasks.
##########################################################################

Reasons for score:

Overall, I vote for rejecting. The idea is a nice contribution but the paper is quite confusingly written and the experiments do not adequately show the impact of the approach. In particular, the deviation between the method and the baseline does not appear to be particularly significant and the authors do not outline their hyperparameter search grid for both method and baselines which makes it quite difficult to tell whether a fair comparison was made. I really enjoy the applications and the idea but feel more careful experimentation is needed to demonstrate that the approach is valuable.
 ##########################################################################

Pros:

1. The paper tackles a practical issue, namely that traffic control architectures are often necessarily decentralized and delayed. This is a good problem to study and a useful add on to existing work studying communication architectures in MARL settings.

2. The idea of adding a predictive forwards model whose output is passed to other agents is a clever way of adding structure to the problem to get around delays. Technically, this does not add anything as an LSTM could internally learn to do this type of prediction as it constructs its messages but structure can often be useful.
3. This partial model based approach may make the controller more robust to changing dynamics conditions or evolving agents.


##########################################################################

Cons:

1. While you tune the value of alpha for your method, it is not clear that you have also given equal amounts of hyperparameter optimization efforts to the baselines. This makes it unclear if your method is actually better or whether you have simply found better hyperparameters.
2. It would be worth expanding on the details of your baselines in the appendix more. It’s not necessary and doesn’t affect my review, but it would make for a better paper!
3. The authors suggest that the world model may “reduce the impacts of nonstationarity” but do not run any experiments that demonstrate this. Additionally, since the world model is implicitly a function of the other agents in the system, the opposite may be true and experiments are not run to investigate this possibility.


A few ablations / changes that might lead to a more insightful paper but are not strictly necessary:
(1) Many of the environments you describe are many-hop and many time-step delayed. It would be interesting to see how your results evolve with longer imagination horizons.
(2) It would be useful for the reader to see the evolution of the world model accuracy over time, to see if it is in fact stationary or convergent. Without this graph, it’s unclear if you are even able to learn a good future predictive model. One way of demonstrating this (though by no means the only way) would be to learn a decoder on the hidden state and see if you can reconstruct the relevant world state variables.
(3) Please add a hyperparameter grid to the appendix so we can see if fair comparison was made to the baselines. Since there do not appear to be existing benchmarks for the baseline methods on these tasks, it is probably necessary to give them equal amounts of tuning effort.

#########################################################################

Things that would improve readability:

- What does it mean to share the policy with the agents around you? What is the representation that is shared?
- The description of environments is a wall of text that is hard to read. Maybe describe the environments at a higher level and move detailed descriptions to the appendix?
- Grammar: grammar throughout the paper is pretty inconsistent and there are too many grammar issues to point them out individually. It may be helpful to pass the paper through an automatic grammar checker and do several more proofreads.
- There is a \varphi in section 4.4 whose purpose I don’t understand; it does not seem to be used anywhere.
- It would be useful to motivate the lower bound that appears in equation (2)
- Including the architecture of the module for a 2-agent case directly in the body of the paper rather than in the appendix would probably make the notation a lot easier to read. As it was, I am still not 100% certain that I understood the notation correctly.
- Figures are small and low-res; I can’t see them very well even by zooming in.

---

> ### Author Response · Authors · 2020-11-19
> **Thanks for your detailed and constructive comments. Here we provide a detailed response to your concerns.**
>
> Thanks for your detailed and constructive comments. Here we provide a detailed response to your concerns. \
> Cons:\
> Q: hyperparameter and the details of your baselines\
> A: Alpha is a tradeoff parameter for the loss of imagination, which is specific to ImagComm.  For the implementation of baselines, we have kept the same setting of hyperparameters, including the dimension of the hidden unit, the learning rates, the usage of recurrent module LSTM for message encoding, which are also consistent with that of NeurComm.
>
> Q: About “reduce the impacts of nonstationarity” \
> A: the nonstationarity comes from the agent’s limited knowledge of the global information. We use predictive communication to enable agents to see more the state of agents that are further away to reduce the nonstationarity.  This is evidenced by the higher reward and lower variance in training.
>
> Ablation/changes:\
> We have updated the submission by including the results of the convergence of the world model in Fig. 7 and the comparison to communication based on 2-hop information in Fig. 8. Here we clarify your concerns.\
> Q: About the world model accuracy over time\
> A: We provide the convergence of the world model in Fig. 7 in appendix. The world model converges well and learns to predict the local transitions of the sender's hidden state.
>
> Q: fair comparison made to the baselines.\
> A: These baselines have previously been considered in NeurComm. All algorithms share the same structure, one fc layer for message   encoding, one LSTM layer for message processing, and hidden unit dimension is set to 64 for all methods. We have used the hyperparameters that reported optimal performance in [1]. The only difference exists in their respective communication scheme.
>
> Readability:\
> Q: What does it mean to share the policy with the agents around you? What is the representation that is shared?\
> A: For the sharing of policy $\pi_i$, we meant the policy fingerprint, which is the probability of action distribution.
>
> Q: There is a $\varphi$ in section 4.4 whose purpose I don’t understand; it does not seem to be used anywhere.\
> A: $\varphi$ denotes the parameters for world model W, as indicated in the end of page 3.
>
> Q: It would be useful to motivate the lower bound that appears in equation (2)\
> A: The motivation is that by maximizing the R.H.S of equation (2), we can gradually increase the value of policy $\pi$ on the left hand side. This is shown in Proposition 2.
>
> [1] Tianshu Chu, Sandeep Chinchali, and Sachin Katti. Multi-agent reinforcement learning for networked
> system control. In International Conference on Learning Representations (ICLR), 2020a.
> URL https://openreview.net/forum?id=Syx7A3NFvH.

---

> > ### Comment · AnonReviewer3 · 2020-11-21
> > **A few more questions**
> >
> > Thanks! So continuing with a few more suggestions:
> > > Q: What does it mean to share the policy with the agents around you? What is the representation that is shared?
> > A: For the sharing of policy \pi, we meant the policy fingerprint, which is the probability of action distribution.
> >
> > Could you include this somewhere in the paper? I'm not seeing it at the moment though I might have missed it. It's important for reproducibility.
> >
> > > A: the nonstationarity comes from the agent’s limited knowledge of the global information. We use predictive communication to enable agents to see more the state of agents that are further away to reduce the nonstationarity. This is evidenced by the higher reward and lower variance in training.
> >
> > I think my issue here is that I don't think non-stationarity is the right term. You're referring to partial observability and pointing out that partial observability leads to non-stationarity in the environment dynamics. I think it might be better to directly make this argument rather than making it implicitly.
> >
> > > A: We provide the convergence of the world model in Fig. 7 in appendix. The world model converges well and learns to predict the local transitions of the sender's hidden state.
> >
> > This is useful! One thing that might make it more useful is to give a sense of what the magnitudes mean. For example, in the first example the loss magnitude is around 1, while in the second it's around 200. How should the reader make sense of those loss magnitudes?
> >
> > > A: These baselines have previously been considered in NeurComm. All algorithms share the same structure, one fc layer for message encoding, one LSTM layer for message processing, and hidden unit dimension is set to 64 for all methods. We have used the hyperparameters that reported optimal performance in [1]. The only difference exists in their respective communication scheme.
> >
> > Please state directly that you're using the hyperparameters for best performance from existing works. However, I assume those hyperparameters were tuned for different environments, right? If my understanding is correct, you can't simply reuse the best hyperparameters, tuned for a different environment, and then use them as a baseline. If this is not the case, I will raise my score but in the absence of it I'm not sure I can raise my score. Benchmarking the performance of your algorithm is really hard and I'm not sure it'd be fair to claim improved performance if the baselines aren't also tuned. Again, I might be misunderstanding you.
> >
> > One last point of concern. I don't think any claims can be made about Fig. 8 because the training clearly hasn't stabilized. This makes it hard to make claims like "our method is better".
> >
> > Finally, I just want to state again that I think this is a really lovely idea but that it might just need a little more focus on either:
> > 1. Showing that it improves SOTA relative to some baselines more carefully
> > 2. More analysis of the method itself to show what is happening in the world model.
> > Either of the two would make this a really good paper!
> >
> > For point 2, I'm not sure just showing the loss of the predictive module is sufficient. It might be worthwhile to actually show the evolution of the predicted quantities versus your predictions.

---

> > > ### Author Response · Authors · 2020-11-22
> > > **Thanks for your follow-up questions, we provide our response below**
> > >
> > > Thanks for your follow-up questions, we provide our response below:\
> > > Q: Regarding the sharing of policy and the impacts of nonstationarity\
> > > A: Thanks for your helpful suggestions, we have corrected the expressions to improve readability.
> > >
> > > Q: Regarding the magnitude in Fig.7\
> > > A: Different from ATSC Grid, the second ATSC Monaco is tasked with heterogeneous agents in which each agent’s state has different dimensions. Thus in (b) we plot the loss that is the summation of the square error of N agents without average or normalization by state dimensions. In (a) the y-axis is averaged RMSE  loss for N agents.
> > >
> > >
> > > Q: Regarding the baselines\
> > > A: Thanks for your acute observation. We have used the same environments/tasks that are used in NeurComm [1], thus reusing the best hyperparameters should not be a concern here.
> > >
> > > Q: Convergence of Fig. 8\
> > > A: Thanks for the acute observation. The variance of CACC tasks is large due to the reward settings;  agents receive a larger penalty of -1000 if they collide with other vehicles. But the conclusion is consistent. ATSC is much more stable, but the training is more time consuming (1-2days for one run) and requires more careful treatment in dealing with multi-hop neighbors in heterogeneous Monaco. We plan to add the results of comparisons on ATSC in Fig. 8 in a future version.
> > >
> > > We sincerely appreciate your suggestions which are very helpful for improving the paper.

---

### Decision · Program_Chairs · 2021-01-07
**Final Decision**

**Decision:**

Reject

**Comment:**

This paper proposes a technique of communicating predicted local states between agents in multi-agent reinforcement learning to deal with the delay in communication.  While the paper addresses an important practical problem, the reviewers have concerns about the insufficiency of novelty and experimental validation.